# Leaky Gut Syndrome Is Associated with Endotoxemia and Serum (1→3)-β-D-Glucan in Severe Dengue Infection

**DOI:** 10.3390/microorganisms9112390

**Published:** 2021-11-19

**Authors:** Wiwat Chancharoenthana, Asada Leelahavanichkul, Wassawon Ariyanon, Somratai Vadcharavivad, Suphasit Phatcharophaswattanakul, Supitcha Kamolratanakul, Pornsawan Leaungwutiwong, Weerapong Phumratanaprapin, Polrat Wilairatana

**Affiliations:** 1Department of Clinical Tropical Medicine, Faculty of Tropical Medicine, Mahidol University, Bangkok 10400, Thailand; supitcha.kam@mahidol.ac.th (S.K.); weerapong.phu@mahidol.ac.th (W.P.); polrat.wil@mahidol.ac.th (P.W.); 2Immunology Unit, Department of Microbiology, Faculty of Medicine, Chulalongkorn University, Bangkok 10330, Thailand; Asada.L@chula.ac.th; 3Translational Research in Inflammation and Immunology Research Unit (TRIRU), Department of Microbiology, Chulalongkorn University, Bangkok 10330, Thailand; 4Cardiometabolic Centre, Department of Medicine, Bangkok Nursing Hospital, Bangkok 10500, Thailand; jowassawon@gmail.com; 5Department of Pharmacy Practice, Faculty of Pharmaceutical Sciences, Chulalongkorn University, Bangkok 10330, Thailand; Somratai.V@chula.ac.th; 6Department of Pharmacy, Hospital for Tropical Disease, Mahidol University, Bangkok 10400, Thailand; suphasit.pha@mahidol.ac.th; 7Department of Microbiology and Immunology, Faculty of Tropical Medicine, Mahidol University, Bangkok 10400, Thailand; pornsawan.lea@mahidol.ac.th

**Keywords:** β-D-glucan, dengue, endotoxins, lactulose-to-mannitol excretion ratio, leaky gut, intestinal permeability

## Abstract

The hallmark of severe dengue infection is the increased vascular permeability and hemodynamic alteration that might be associated with an intestinal permeability defect. However, the mechanisms underlying the gastrointestinal-related symptoms of dengue are not well characterized. A prospective observational study was conducted on patients with dengue who were categorized according to: (i) febrile versus critical phase and (ii) hospitalized patients with versus without the warning signs to evaluate the gut barrier using lactulose-to-mannitol excretion ratio (LEMR). Serum endotoxins, (1→3)-β-D-glucan (BG), and inflammatory parameters were measured. A total of 48 and 38 patients were enrolled in febrile illness and critical phase, respectively, while 22 and 64 patients presented with or without the warning signs, respectively. At enrollment, a positive LEMR test was found in 20 patients (91%) with warning signs, regardless of phase of infection. Likewise, serum endotoxins and BG, the indirect biomarkers for leaky gut, prominently increased in patients who developed severe dengue when compared with the non-severe dengue (endotoxins, 399.1 versus 143.4 pg/mL (*p* < 0.0001); BG, 123 versus 73.8 pg/mL (*p* = 0.016)). Modest impaired intestinal permeability occurred in dengue patients, particularly those with warning signs, and were associated with endotoxemia and elevated BG. Thus, leaky gut syndrome might be associated with severity of dengue infection.

## 1. Introduction

Dengue virus (DENV), the most prevalent mosquito-borne viral pathogen among humans globally, was identified by the World Health Organization (WHO) as one of the top 10 threats to global health in 2019 [1]. An estimated annual incidence of dengue viral infection has increased progressively over the last 50 years [2]; approximately 40% of the world is at risk of dengue fever, and there are approximately 390 million infections a year [1]. The latest data showed that the trajectory of annual incidence of dengue infection by using climate variability, population, and socioeconomic projection was approximately 60% of the global population by the year 2080 [3]. This data set is more accurate than that analyzed in a previous study because dengue infection was used as the definition, not dengue fever, which may include atypical presentation in some individuals. Accordingly, WHO’s dengue control strategy aims to reduce deaths by 50% by 2020 [1]. Although most infected individuals develop only benign febrile illness (4–7 days of self-limited fever), some patients develop a life-threatening syndrome, also known as severe dengue infection or dengue hemorrhagic fever (DHF). As such, progression to severe dengue commonly occurs after the febrile phase, i.e., between days 3 and 7 of the onset [4], and warning signs that might be associated with the development of severe dengue can be observed. Severe dengue has potentially fatal complications, including plasma leakage, fluid accumulation, respiratory distress, severe bleeding, or organ impairment. The prediction of severe dengue during the febrile phase and appropriately prompt interventions are key to virus management and to reducing the mortality rate, which will decrease the societal healthcare burden of dengue.

However, the mechanisms underlying the gastrointestinal-related symptoms of dengue remain poorly understood. In 2009, the WHO announced a classification highlighting a set of warning signs, including abdominal pain or tenderness, persistent vomiting, clinical fluid accumulation, mucosal bleeding, lethargy or restlessness, liver enlargement >2 cm, and an increase in hematocrit concurrent with a rapid decrease in platelet count [4], to help identify patients likely to progress to severe dengue. Indeed, a substantial proportion of patients with dengue infection demonstrate gastrointestinal symptoms, including nausea, vomiting, and loss of appetite, and most patients develop some degree of liver injury and increased vascular permeability, particularly in severe dengue [5].

Although enterocyte injuries in severe dengue infection are mainly due to intestinal injury from plasma leakage-induced hemodynamic alteration (dengue shock syndrome), increased serum lipopolysaccharide (LPS or endotoxin) is correlated with dengue severity [6,7,8,9]. Because endotoxin is a major component of Gram-negative bacteria (the most predominant gut organisms), the detection of LPS in serum without Gram-negative bacterial infection is an indirect indicator of gut barrier defects [6,7,8,9]. Additionally, LPS is not only a potent activator of inflammatory pathways through several interactions, including LPS-binding protein (LBP), soluble cluster of differentiation 14 (sCD14), and Toll-like receptor 4 (TLR4), on monocytes and macrophages [10] but also acts synergistically with dengue virus to induce the production of platelet activating factor (PAF) and other inflammatory cytokines, including tumor necrosis factor (TNF)-α and interleukin (IL)-1β, which could contribute to dengue disease severity [11]. Because gut fungi, especially *Candida albicans* in humans, are the second most abundant organisms in gut contents, an elevation of serum (1→3)-β-D-glucan (BG), a major component of the fungal cell wall, without systemic fungal infection is also proposed as an indirect parameter of gut barrier defects [12], similar to serum LPS [6,7,8,9]. Similarly, BG also activates immune responses through several pathways, especially Dectin-1, which could enhance LPS immune responses partly through the cross-link between TLR-4 and Dectin-1 [12]. As such, leaky gut syndrome, also known as increased intestinal permeability, is a digestive condition in which bacteria and toxins can translocate into blood circulation and are associated with several medical conditions, such as bacterial sepsis [13], hepatic disease [14], and lupus [15]. Although the gold standard intestinal permeability test is the detection of disaccharides (lactulose) that cannot be absorbed through intestines in urine after oral administration [16], studies of leaky gut using this standard method are still limited. Due to the endogenous sources of LPS and BG gastrointestinal tracks, we hypothesized that endotoxemia (LPS) and BG in the blood of patients with dengue are most likely responsible for gut barrier defects (leaky gut syndrome).

Because (i) leaky gut in dengue might be another early marker of dengue disease progression and (ii) the proinflammatory effect of LPS and BG in dengue might enhance dengue severity, an initial exploration of leaky gut on dengue is warranted before further clinical use. Therefore, the intestinal permeability in patients with dengue between the group with and without warning signs was explored. Moreover, the use of the lactulose–mannitol excretion ratio in urine, together with endotoxemia, serum BG, and inflammatory serological markers was also explored.

## 2. Materials and Methods

### 2.1. Enrolled Participants and Study Designs

We conducted a prospective cohort study between November 2020 and September 2021. The protocol was approved by the Ethics Committee of the Faculty of Tropical Medicine, Mahidol University (MUTM 2020-071-01), and registered in the Thai Clinical Trials Registry (TCTR20210208002) in accordance with STROBE guidelines, and written informed consent was obtained from all participants.

To exclude age-related physiologic alterations in intestinal permeability, patients 18–50 years old were recruited. Dengue was diagnosed by the NS1 antigen test (Platelia enzyme-linked immunosorbent assay (ELSA); Bio-Rad) and commercial immunoglobulin (Ig) M and IgG serology assays (Capture ELISA; Panbio) during the febrile or critical phase. In addition, reverse-transcription polymerase chain reaction (RT–PCR) was performed on enrollment samples to identify the viral serotype. Patients were defined as having laboratory-confirmed dengue if the RT–PCR, NS1 antigen, or IgM assays were positive at enrollment or if there was IgM seroconversion between the paired specimens, following World Health Organization (WHO, Geneva, Switzerland) criteria [17]. Sample size was determined based on a previous endotoxin study [8]. From the mean difference of 0.705, the power calculation showed that a total sample size of 79 was required to achieve 80% power for detection of a statistically significant effect size with an alpha error of 0.05. The exclusion criteria were severe dengue clinical presentation at enrollment; reported recent use of medication or over the counter medication (e.g., prebiotic, probiotic, or vitamin supplements); history of gastrointestinal surgery; family history of inflammatory/irritable bowel syndrome; hypersensitivity reaction to mannitol and/or lactulose; presence of alcohol intake or substance misuse; pregnancy; menstruation period; and an inability to speak Thai. The use of medication other than acetaminophen was prohibited during the study period. All patients were reviewed daily until fully recovered and afebrile or for up to 7 days after enrollment.

Participants were tested for intestinal permeability with both the urinary lactulose-to-mannitol excretion ratio (LMER) test and plasma biomarkers. Indeed, the urinary LMER was tested only day +1 after admission (equivalent to date of enrollment, *t* = 0), at which time the severity of dengue infection remained unknown. Several serum parameters, including (1→3)-β-D glucan (BG), lipopolysaccharide (LPS), LPS-binding protein (LBP), and inflammatory cytokines, including interleukin (IL)-6, IL-8, IL-1β, and tumor necrosis factor (TNF)-α cytokines, were tested on both day +1 of admission and day +7 of fever onset. In addition, healthy men and women (*n* = 5) were recruited as a control group for LMER reference. The detailed procedure of the study design is depicted in Figure 1A, whereas the timeline of the study is shown in Figure 1B.

### 2.2. Definition

Intact intestinal permeability refers to a lactulose-to-mannitol excretion ratio > 0.1178, while impaired intestinal permeability means an LMER ≤ 0.1178 (see below). According to the 2009 WHO guideline for dengue infection [16], the febrile phase of dengue infection refers to days 1–3 of illness, and the critical phase is the time between days 4 and 7 of illness. The warning signs consist of abdominal pain or tenderness, persistent vomiting, clinical fluid accumulation, mucosal bleeding, lethargy or restlessness, liver enlargement > 2 cm, and increase in hematocrit concurrent with rapid decrease in platelet count. In parallel, severe dengue infection refers to dengue patients with at least one of the following symptoms: (i) severe plasma leakage, (ii) severe bleeding, and (iii) severe organ involvement [16].

### 2.3. Gut Permeability Analysis

The mixture of lactulose and mannitol consisting of 5 g lactulose, 2 g mannitol, and 22.3 g glucose (as an osmotic filler) dissolved in 100 mL sterile water with osmolarity at 1500 mOsmol/L was produced from the Faculty of Tropical Medicine, Mahidol University, and used as approved by the Ethics Committee that was previously mentioned. To evaluate intestinal permeability, the primary outcome, the urinary lactulose-to-mannitol excretion ratio (LMER), was assessed. Isocratic ion-exchange high-performance liquid chromatography (HPLC) with mass spectrometry (Thermo Fischer Scientific, Waltham, MA, USA) was used to determine sugar concentrations in urine samples following a previous publication [18]. Briefly, an oral administration of lactulose and mannitol mixtures followed by a time-point urine collection for the subsequent 5 h was collected with 1 h separated duration, including urine collection between 0 and 1 h, 1 and 2 h, 2 and 3 h, 3 and 4 h, and 4 and 5 h after administration. No food was allowed during the urine collection period. Because the molecular size of lactulose is too large to pass through the gut barrier, the detection of lactulose in urine indicates intestinal tight junction injury [16]. Additionally, mannitol is small enough to transcellularly cross the gut barrier and is used as a control molecule that might be influenced by dilution of the mixtures, gastric emptying time, transit time, epithelial absorptive area, and renal function [16,19]. Since the distribution of LMER in the validation phase was normally distributed by the D’Agostino and Pearson test, the values are presented as the mean ± standard deviation (SD), and LMER in healthy volunteers (*n* = 20, 50% was female) in our center was 0.04 (range 0.02–1.71) and 10.44 (range 1.68–35.60), respectively.

### 2.4. Serum Parameters

Samples were used to measure LPS-binding protein (LBP), and cytokines (IL-6, IL-8, IL-1β, and TNF-α) were measured using ELISA from Hycult Biotech, Uden, The Netherlands, and Meso Scale Discovery, Rockville, MA, USA. Serum BG was measured by a Fungitell^®^ assay (Associates of Cape Cod, Falmouth, MA, USA) that detects BG through activation of factor G, a protease zymogen, which cleaves a chromophore from a chromogenic peptide through light absorbance at 405 nm. Briefly, 5 mL of sample was mixed with an alkaline pretreatment reagent (0.125 M KOK/0.6 M KCl) and incubated at 37 °C for 10 min, and reconstituted Fungitell reagent (100 µL) was added to each well before measurement. Because the detectable range of BG was 7.8–523.4 pg/mL, BG values of <7.8 pg/mL and >523.4 pg/mL were recorded as 0 and 523 pg/mL, respectively. In parallel, serum lipopolysaccharide (LPS) was assessed by chemiluminescence-based endotoxin activity (EA) assay (Spectral Diagnostics, Toronto, ON, Canada) based on the detection of enhanced respiratory burst activity in neutrophils following their priming by complexes of endotoxin and a specific anti-endotoxin antibody [20,21]. Briefly, whole blood (50 μL) was incubated in duplicate with saturating concentrations of an anti-lipid A IgM antibody and stimulated with opsonized zymosan, and respiratory burst activity was detected from the lumiphor luminol by a chemiluminometer (Berthold Technologies, Bad Wildbad, Germany). By measuring basal and maximally stimulated responses, EA levels can be expressed in relative units derived from the integral of the basal and stimulated chemiluminescent responses. The endotoxin results in EU/mL were converted to picograms per milliliter (pg/mL) and log transformed to standardize the data distribution to a log-normal distribution. As such, EA levels were classified as “low” (0.0 to 0.39), “intermediate” (0.40 to 0.59), and “high” (≥0.6), and EA levels of 0.4 and 0.6 are approximately equivalent to endotoxin concentrations of 500 and 1500 pg/mL, respectively. Most of the parameters were measured on the admission date (*t* = 0) and on the seventh day of admission (*t* = 7), as shown in Figure 1. All samples were kept frozen at −80 °C until analyses.

### 2.5. Statistical Analysis

Quantitative data are summarized as the mean and standard deviation (SD) for normally distributed variables or median (IQR) for nonnormally distributed variables, and qualitative data are presented as *n* (percentage). Normally distributed variables, nonnormally distributed variables, and categorical variables were compared by Student’s *t* test, Mann–Whitney U test, and chi-square test, respectively. Correlations were calculated using Pearson’s correlation coefficient. Logarithmic transformation was used if the data were not normally distributed. A *p* < 0.05 was considered statistically significant. Data analysis was performed using the PASW 18.0.0 statistical software package (SPSS Inc., Chicago, IL, USA). The receiver operating characteristic (ROC) curves of cytokine response and figures were made using GraphPad Prism 9.2.0 software (GraphPad Software, Inc., La Jolla, CA, USA).

## 3. Results

### 3.1. Baseline Characteristics of Participants

A total of 86 dengue inpatient and 5 healthy control participants were included in the study. Of all patients, 48 (55.8%) patients were in the acute phase of dengue infection, whereas 38 (44.2%) patients were in the convalescent phase of dengue infection. The baseline characteristics of patients with dengue in both groups were similar in terms of demographics and comorbidities but not gastrointestinal symptoms, hemoconcentration, or platelet counts (Table 1). Of these, 41 (64%) dengue patients without warning signs had at least one gastrointestinal symptom, but all dengue patients with warning signs had more than one gastrointestinal symptom. All participants fully recovered at the end of the observation without dropping out. Using the 2009 WHO classification, there were 22 patients (25.6%) with dengue warning signs and 64 patients (74.4%) with dengue without warning signs. Among 74 patients with PCR positivity for dengue, the number of patients with the dengue virus serotypes DENV-1 and DENV-2 was 23 (31.1%) and 35 (47.3%), respectively, and 16 patients (21.6%) had DENV-4. In addition, all baseline intestinal permeability and inflammatory marker data, but not lipopolysaccharide-binding protein (LBP), were significantly different among the groups, as shown in Table 1.

### 3.2. Comparisons of Intestinal Permeability among Groups Using the Lactulose-to-Mannitol Excretion Ratio (LMER)

Dengue patients with and without warning signs all underwent intestinal permeability tests using LMER. In terms of mannitol excretion, there was no significant difference among the groups, implying nondifferent gastrointestinal absorption through the intestinal transcellular pathway (Figure 2A). In parallel, the healthy control subjects showed significantly lower lactulose excretion and LMER than dengue patients with warning signs (*p* < 0.0001), and the LMER was significantly different among all groups (Figure 2A,B). Of note, 20 of 22 participants (91%) in the dengue with warning signs group had positive LMER (impaired intestinal permeability), whereas only 11 of 64 participants (17%) in the dengue without warning signs group had positive LMER. Moreover, 8 (16.7%) patients were in the acute phase of infection, compared with 40 (83.3%) patients with negative LMER (intact intestinal permeability). Similarly, most dengue patients who had warning signs in the convalescent phase showed positive LMER (Table 2).

### 3.3. The Progression to Severe Dengue Infection as Stratified by the Lactulose-to-Mannitol Excretion Ratio (LMER) Test and Warning Signs of Dengue Infection

On the seventh day of admission (*t* = 7) (Figure 1B), there were 27 of 86 patients (31.4%) with severe dengue infection. The severe dengue infection characteristics were (i) severe plasma leakage leading to shock in 14 patients (51.9%) or fluid accumulation with respiratory distress in 4 patients (14.8%); (ii) severe bleeding in 4 patients (14.8%); and (iii) severe organ impairment as aspartate transaminase (AST) > 1000 U/L in 5 patients (18.5%). On the other hand, the characteristics of 27 patients with severe dengue as categorized by LMER and warning signs were (i) 18 patients (90%) with positive LMER plus positive warning signs (2 and 16 patients from acute and convalescent phases of infection, respectively), (ii) 6 patients (54.5%) with positive LMER without warning signs (3 patients from acute phase and another 3 patients from convalescent phase of infection), and (iii) 3 patients (5.7%) with negative LMER without warning signs (2 patients from acute phase and 1 patient from convalescent phase of infection).

### 3.4. Intestinal Damage, Lipopolysaccharides (LPS), and Inflammatory Markers

Next, we investigated intestinal damage and inflammatory markers. Circulating LPS-binding protein (LBP) and (1→3)-β-D-glucan (BG) were therefore assessed and stratified by the final outcomes of dengue infection—severe and nonsevere dengue infection cases—with retrospective analysis. Although LBP levels showed a nonsignificant difference between patients with severe versus nonsevere dengue infection at enrollment (*t* = 0), lower LBP levels were found in severe dengue patients than in nonsevere dengue patients (*p* = 0.02) (Figure 3A). In contrast, serum BG levels were higher in severe dengue patients than in nonsevere patients and healthy controls (*p* = 0.02 and *p* < 0.0001, respectively) at enrollment (*t* = 0), and the difference was more prominent on day 7 of the observation (*t* = 7) (Figure 3B).

Additionally, proinflammatory cytokines, including interleukin (IL)-6, IL-8, IL-1β, and tumor necrosis factor (TNF)-α, at enrollment (*t* = 0) were not different between the severe versus nonsevere dengue infection groups, and these cytokines were significantly higher than those in heathy control subjects (Figure 3C). Likewise, these cytokines were higher in severe dengue infection patients than in nonsevere dengue infection patients and healthy control subjects at day 7 (*t* = 7) (Figure 3D).

Regarding endotoxemia, dengue patients with and without warning signs presented significantly higher serum LPS concentrations at enrollment than healthy control subjects (*p* < 0.0001) (Figure 3E), and serum LPS was higher in patients who progressed to severe dengue infection but not in those with nonsevere infection (Figure 3E). Of note, 20 patients (74%) with severe dengue infection received empirical antibiotics preceding the second serum LPS evaluation. With subgroup analysis using the presence of warning signs, most patients with severe dengue infection demonstrated positive LMER together with warning signs at enrollment (Figure 3F).

### 3.5. Increased Lipopolysaccharides (LPS) and (1→3)-β-D-Glucan (BG) in Serum in Parallel with a Positive LMER Test in Patients with Dengue

Because of the proposed indirect parameters of gut barrier defects using serum LPS and BG due to the more convenient measurement than the LMER assay, the correlation between LPS and BG in serum versus intestinal permeability test by LMER was conducted. Indeed, both serum LPS and BG showed a positive correlation with increased LMER, but serum LPS had a better correlation with LMER (r^2^ = 0.562, *p* < 0.0001) than BG (r^2^ = 0.189, *p* = 0.045) (Figure 4A,B). Receiver operating characteristic (ROC) curves of circulating LPS and BG levels were also performed to determine the cutoff point of gut barrier defects that indicated a possible severe dengue infection. Accordingly, the cutoff levels of LPS and BG for severe dengue prediction were 238.5 pg/mL (sensitivity 74.1% and specificity 95%) and 59.5 pg/mL (sensitivity 85% and specificity 57%), respectively. Additionally, the cutoff levels of LPS and BG for patients who underwent severe dengue at the end were 149.5 pg/mL (sensitivity 63% and specificity 71%) and 103 pg/mL (sensitivity 74% and specificity 90%), respectively (Figure 4C,D). Notably, 64% of participants in this cohort received empirical antibiotics for any reason during the observation that might have interfered with leaky gut. Interestingly, LMER of 0.144 was a fair predictor for severe dengue severity, with a sensitivity of 53% and specificity of 71% (Table 3).

## 4. Discussion

Although serum (1→3)-β-D-glucan (BG) and lipopolysaccharides (LPS) have been proposed as indirect biomarkers of gut barrier defects in several publications [12,13,22,23,24], the association between these biomarkers and a standard gut permeability test in patients was clearly observed for the first time herein. Despite a better correlation between serum LPS and gut barrier defects than serum BG, both LPS and BG were correlated with gut permeability injury, which may be useful for the prediction of severe dengue infection.

Endotoxin (LPS) is a major component of Gram-negative bacteria in the gut [25], and the presence of LPS in circulation without bacterial infection implies microbial gut translocation [26]. Here, serum endotoxin was demonstrated in both severe and nonsevere dengue infection, but the level was higher in severe infection (Figure 3E), supporting several previous publications possibly due to leaky gut [6,7,8,9]. Interestingly, the intestinal permeability test using the fractional excretion of lactulose–mannitol in urine after oral administration (lactulose-to-mannitol excretion ratio, LMER), a gold standard test for the gut barrier [27] in several clinical studies (such as Crohn’s disease) [27,28,29,30], was well correlated with serum LPS, supporting the use of LPS as a biomarker for the gut barrier in patients with dengue. Because the LMER test could not be performed in patients with severe dengue because of the limited oral intake during the standard dengue treatment, the LMER test data in severe dengue were very limited. However, LMER was a potential test for predicting severe dengue infection with LEMR higher than 0.1444 (sensitivity 58% and specificity 71%), which was mostly found in dengue with positive warning signs (predominantly in the critical phase of infection) (Table 2). Additionally, all dengue patients with warning signs had more than three gastrointestinal symptoms compared with zero to two symptoms in dengue without warning signs; therefore, impaired intestinal permeability (as indicated by a high LMER value) might induce gastrointestinal symptoms, and the LMER test might be helpful for the prediction of dengue severity. Perhaps dengue virus not only directly leads to gut leakage by affecting intestinal wall integrity through alterations of epithelial cells and tight junctions causing epithelial cell damage [31] but also indirectly affects intestinal barriers by altering the cellular microenvironment [32].

Because (1→3)-β-D-glucan (BG) is a major component of gut fungi, the second most predominant gut organism, the detection of serum BG has also been proposed as an indirect biomarker of gut barrier defects [33,34]. Indeed, serum BG showed better differentiation between severe and nonsevere dengue infection in both the febrile and critical phases (Figure 3B) than LPS (Figure 3E). Nevertheless, LPS levels might be affected by antibacterial administration during the course of treatment. Nevertheless, with our current limitation, serum BG levels outperformed LPS in terms of diagnostic markers for severe dengue infection but not for a prediction marker (Table 3). Although BG is typically recognized as an innate immune system-activating pathogen-associated molecular pattern (PAMP) and influences innate immune metabolism [35], BG is dependent upon myriad factors, including host immune response and clearance efficacy [36].

Because both LPS and BG are PAMPs that could prominently activate immune responses, these PAMPs from gut translocation might enhance systemic inflammation and worsen dengue severity. As such, lipopolysaccharide (LPS)-binding protein (LBP) plays a crucial role in the innate immune response for the development of inflammatory and infectious-related diseases that are produced by hepatocytes in response to proinflammatory cytokines (IL-6, IL-1β, and TNF-α) [37], as demonstrated in patients with bacterial and fungal sepsis [38]. Here, LBP levels in a viral infection were also higher in patients with dengue infection than in controls but were not different between patients with severe and nonsevere dengue infection (*t* = 0). Interestingly, LBP was significantly lower during the follow-up, especially on the seventh day of the observation (*t* = 7), particularly in patients with severe dengue infection (Figure 3A), possibly due to a very short half-life (as a few days) of LBP [39] and hepatocyte exhaustion (liver congestion) [8]. Taken together, as an acute phase protein, the LBP level is not suitable as a prognostic marker for severe dengue infection. Notably, increased LBP showed a similar pattern as LPS levels in the febrile phase, likely due to the LPS neutralization effect of LBP [40]. On the other hand, all proinflammatory cytokines (IL-6, IL-8, IL-1β, and TNF-α) in patients were higher than those in the control group, partly due to TLR4 activation by LPS [41]. Therefore, in the future, determination of TNF-α receptor levels could clarify the involvement of TNF-α.

Taken together, we propose the use of BG and LPS as surrogate markers for predicting and confirming gut leakage-associated severe dengue infection in patients who have either warning signs or at least three gastrointestinal symptoms, particularly in the case of those signs and symptoms that are found in the critical phase. Cost–benefit analysis should be addressed in further studies. This study, however, had several limitations. First, only 64% of participants in this cohort were assessed for blood bacterial isolation and received empirical antibiotics due to a suspicion of cobacterial infection. Because the endotoxin levels detected by the endotoxin activity assay in our patients were mostly higher than 50 pg/mL (the levels of Gram-negative bacterial infection) [42] without profound sources of bacterial infection, the high-level serum LPS in our patients should be responsible for the nonbacterial infection. In addition, none of the participants showed positive bacterial isolation. Second, there was no mortality in this cohort. Therefore, the predictive value of LMER test, LPS, and BG levels for mortality could not be concluded. Third, owing to the symptoms mimicking sepsis, confounding effects from antibiotic treatment may have existed between groups, leading to the underestimation of LPS concentration. Finally, severe dengue infection patients with shock or hemorrhage were not enrolled because we were unable to study the LMER test in severe dengue patients. As such, the results could not be generalized for all severities of dengue infection. The present study excluded dengue patients who were younger than 18 years old, which is an important age group prone to dengue infection. We were also unable to study elderly adults, the age group with the greatest risk of severe dengue and poor outcomes [43], who might have more profound gut barrier damage.

In summary, our study has highlighted endotoxemia and serum BG as indirect biomarkers for impaired intestinal permeability tests (leaky gut syndrome) that might be useful for the prediction of dengue disease severity. Although targeted restoration of intestinal barrier dysfunction seems to be a logical step in the modulation of either intestinal diseases or systemic diseases, there is currently no proper therapy for gut barrier improvement for clinical use in dengue. The increased understanding of the potential related mechanisms that worsen dengue infection from the present study is likely to provide novel approaches for disease modulation. Further studies are necessary.

## Figures and Tables

**Figure 1 microorganisms-09-02390-f001:**
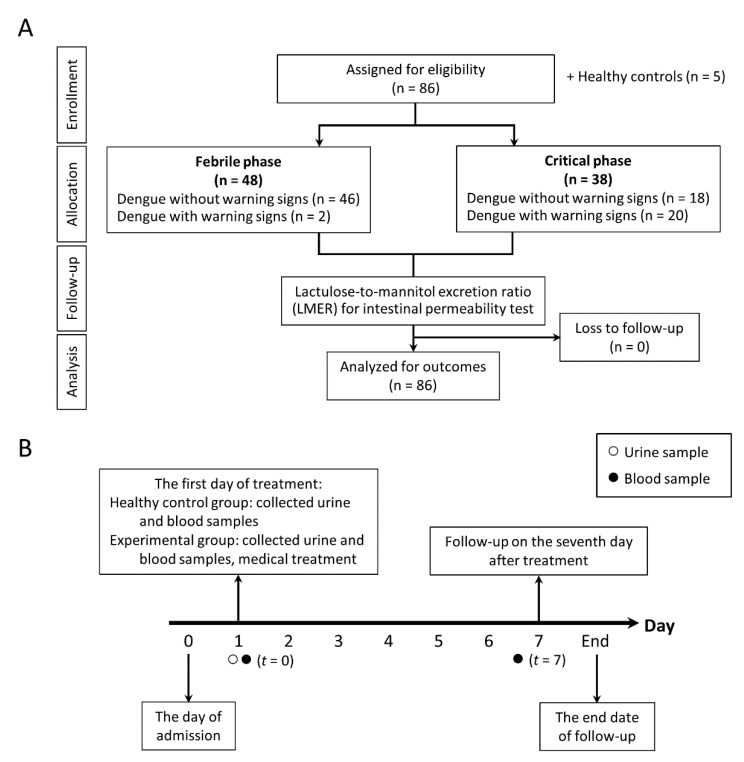
The schematic diagram represents (**A**) study flow diagram and (**B**) timeline of the study design.

**Figure 2 microorganisms-09-02390-f002:**
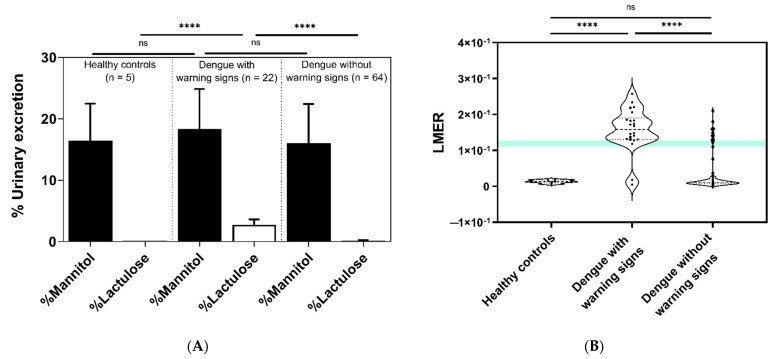
Urinary excretion of mannitol and lactulose from participants among three groups (healthy control subjects (*n* = 5) and dengue patients with (*n* = 22) or without warning signs (*n* = 64)). (**A**) The percentages (%) of urinary mannitol and lactulose excretion demonstrated significantly higher lactulose excretion in dengue patients with warning signs than in those without warning signs, but there was no significant difference in mannitol excretion among the groups. (**B**) Comparisons of the lactulose-to-mannitol excretion ratio (LMER) demonstrated by the violin plot again demonstrated a significantly higher LMER in dengue patients with warning signs than in dengue patients without warning signs (20 from 22 versus 11 from 64 participants). The transverse dotted line demonstrates the median and quartiles. A dot refers to an individual value. The light blue area indicates the 90th percentile of the upper normal limits (0.1187) as a reference and serves as the cutoff value for a positive LMER test. (**** *p* < 0.0001 and ns stands for nonsignificant.)

**Figure 3 microorganisms-09-02390-f003:**
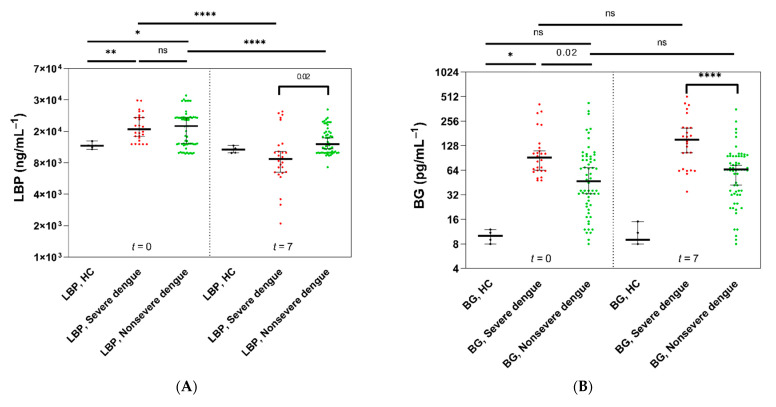
Ranges of measured lipopolysaccharide-binding protein (LBP), (1→3)-β-D-glucan (BG), cytokines, and lipopolysaccharides (LPS). The detection range of tested serum samples on 1 (*t* = 0) and 7 (*t* = 7) of admission from patients who developed severe dengue (red, *n* = 27) and nonsevere dengue (green, *n* = 59) in comparison with serum samples from healthy control subjects (black, *n* = 5) is demonstrated. Heavy bars indicate medians, and error bars represent 95% CIs. Each value is indicated by a dot. Log transforms with nonparametric statistics were performed as skewed data. (**A**) demonstrates a tendency of higher LBP in patients who underwent severe dengue infection compared with those with nonsevere dengue and healthy control subjects at enrollment (significantly decreased LBP in severe dengue patients on day 7). (**B**) demonstrates significantly increased BG levels in severe dengue patients compared with nonsevere dengue patients. (**C**,**D**) demonstrate significantly higher levels of interleukin (IL)-6, IL-8, IL-1β, and TNF-α in severe dengue patients than in nonsevere dengue patients and healthy control subjects. (**E**) demonstrates significantly increased circulating LPS in severe dengue infection at enrollment before antibiotic treatment. (**F**) demonstrates that most severe dengue patients had a positive lactulose-to-mannitol excretion ratio (LMER) corresponding to the presence of warning signs at enrollment. (**** *p* < 0.0001, *** *p* < 0.001, ** *p* < 0.01, and * *p* < 0.05).

**Figure 4 microorganisms-09-02390-f004:**
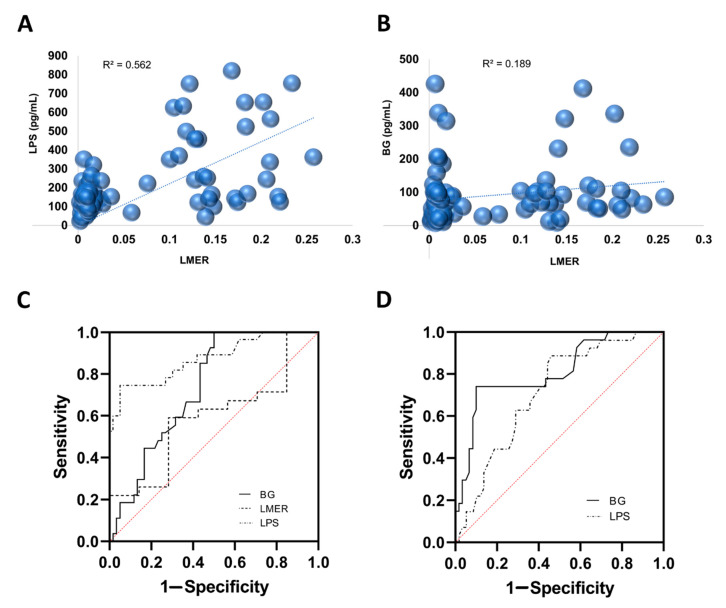
The relationship between the lactulose-to-mannitol excretion ratio (LMER) and serum lipopolysaccharides (LPS) and (1→3)-β-D-glucan (BG). (**A**) Increased LMER demonstrates a better correlation with serum LPS than (**B**) BG. The ROC (receiver operating characteristic) curves for (**C**) the severe dengue prediction of BG, LMER, and LPS and for (**D**) the severe dengue infection confirmation from BG, LMER, and LPS are shown.

**Table 1 microorganisms-09-02390-t001:** Differences in baseline characteristics of dengue patients with and without warning signs, according to the World Health Organization 2009 classification.

	Control Group (*n* = 5)	Dengue with Warning Signs (*n* = 22)	Dengue without Warning Signs (*n* = 64)	*p* Value ^1^
Age, years	36.4 ± 8.2	41.4 ± 10.6	40.1 ± 9.8	0.60
Female, *n* (%)	3 (60.0)	16 (72.7)	44 (68.8)	0.73
Day of illness at enrollment		5.2 ± 0.4	3.4 ± 0.5	<0.0001
NS1 positive, *n* (%)		22 (100.0)	64 (100.0)	-
PCR positive, *n* (%)		22 (100.0)	52 (81.3)	0.03
Serotype DENV-1		1 (4.5)	22 (42.3)	0.001
Serotype DENV-2		15 (68.2)	20 (38.5)	0.02
Serotype DENV-3		0 (0)	0 (0)	-
Serotype DENV-4		6 (27.3)	10 (19.2)	0.43
Comorbidities, *n* (%)		5 (22.7)	11 (17.2)	0.57
Diabetes		1 (4.5)	3 (4.7)	0.97
Hypertension		5 (22.7)	10 (15.6)	0.45
Symptoms, *n* (%)				
Fever		22 (100.0)	64 (100.0)	-
Anorexia		22 (100.0)	55 (85.9)	0.06
Nausea		22 (100.0)	41 (64.1)	0.001
Vomiting		12 (54.5)	0 (0)	<0.0001
Abdominal pain		19 (86.4)	0 (0)	<0.0001
Diarrhea		7 (31.8)	0 (0)	<0.0001
Myalgia		22 (100.0)	14 (21.9)	<0.0001
Rash		11 (50.0)	16 (25.0)	0.03
Mucosal bleeding		7 (31.8)	0 (0)	<0.0001
Clinical fluid accumulation		5 (22.7)	0 (0)	0.0001
Hemoglobin (g/dL)		14.6 ± 0.9	13.3 ± 1.1	<0.0001
Hematocrit (%)		43.1 ± 2.2	40.2 ± 0.7	<0.0001
White blood cell count (×10^9^/L)		3.83 ± 1.4	5.28 ± 2.1	0.003
Platelet count (×10^9^/L)		93.2 ± 11.6	121.2 ± 14.5	<0.0001
Albumin (g/L)		40.8 ± 1.5	41.8 ± 2.2	0.05
Aspartate transaminase (AST) (U/L)		134.5 ± 16.2	68.8 ± 10.7	<0.0001
Alanine transaminase (ALT) (U/L)		50.1 ± 6.9	41.9 ± 10.4	0.001
Positive cases of bacterial isolation from blood, n (%) ^2^		0 (0)	0 (0)	-
LMER, median (range)	0.013 (0.01)	0.158 (0.25)	0.011 (0.21)	<0.0001
Endotoxins (pg/mL)	33.4 ± 10.1	332.8 ± 263.2	103.4 ± 66.3	<0.0001
Lipopolysaccharide-binding protein (LBP) (ng/mL)	10,070 ± 109.6	15,550 ± 4916.4	16,336 ± 7246.8	0.64
(1→3)-β-D-glucan (BG) (pg/mL)	10.0 ± 1.6	103.0 ± 96.9	53.8 ± 33.1	0.0007
IL-6 (pg/mL)	24.2 ± 10.8	408.0 ± 203.5	202.7 ± 146.3	<0.0001
IL-8 (pg/mL)	78.6 ± 15.7	772.5 ± 330.5	192.3 ± 109.3	<0.0001
IL-1β (pg/mL)	18.6 ± 8.3	215.4 ± 147.3	78.8 ± 68.0	<0.0001
TNF-α (pg/mL)	14.4 ± 4.0	342.8 ± 235.4	206.5 ± 130.5	0.001

^1.^ Comparison between dengue with and without warning signs. ^2^ Data from 64% of patients who received empirical antibiotics. IL, interleukin; LMER, lactulose-to-mannitol excretion ratio; TNF, tumor necrosis factor.

**Table 2 microorganisms-09-02390-t002:** The outcomes of the intestinal permeability test by using the lactulose-to-mannitol excretion ratio (LMER) stratified by warning signs and phases of dengue infection at the time of enrollment (*t* = 0).

	Febrile Phase		Critical Phase	
	Dengue with Warning Signs (*n* = 2)	Dengue without Warning Signs (*n* = 46)	Dengue with Warning Signs (*n* = 20)	Dengue without Warning Signs (*n* = 18)
Positive LMER test, *n* (%)	2 (100.0)	6 (13.0)	18 (90.0)	5 (27.8)
Negative LMER test, *n* (%)	0 (0)	40 (87.0)	2 (10.0)	13 (72.2)

**Table 3 microorganisms-09-02390-t003:** Parameters calculated by ROC curves of the lactulose-to-mannitol excretion ratio (LMER), lipopolysaccharide (LPS), and (1→3)-β-D-glucan (BG) for severe dengue prediction and diagnosis.

For Severe Dengue Infection Prediction					
Cutoff	Sensitivity (%)	95% CI ^1^	Specificity (%)	95% CI	Likelihood Ratio
LMER					
0.1382	62.5	43–79%	42.9	16–75%	1.10
0.1401	62.5	43–79%	57.1	25–84%	1.46
0.1420	58.3	39–76%	57.1	25–84%	1.36
0.1444	58.3	39–76%	71.4	36–95%	2.04
0.1470	54.2	35–72%	71.4	36–95%	1.90
0.1580	50.0	31–69%	71.4	36–95%	1.75
0.1698	45.8	28–65%	71.4	36–95%	1.60
LPS (pg/mL)					
233.0	74.1	55–87%	89.8	80–95%	7.28
235.5	74.1	55–87%	91.5	82–96%	8.74
238.5	74.1	55–87%	94.9	86–99%	14.57
242.5	70.4	52–84%	94.9	86–99%	13.84
247.5	66.7	48–81%	94.9	86–99%	13.11
255.0	63.0	44–78%	94.9	86–99%	12.38
261.5	59.3	41–75%	94.9	86–99%	11.65
BG (pg/mL)					
50.0	93.0	77–99%	51.7	39–64%	1.92
51.5	88.9	72–96%	53.3	41–65%	1.91
54.0	85.2	68–94%	53.3	41–65%	1.83
57.0	85.2	68–94%	55.0	42–67%	1.90
59.5	85.2	68–94%	57.0	44–68%	1.97
62.0	81.4	63–92%	57.0	44–68%	1.88
63.5	74.1	55–87%	57.0	44–68%	1.71
**For Severe Dengue Infection Diagnosis**					
**Cutoff**	**Sensitivity (%)**	**95% CI**	**Specificity (%)**	**95% CI**	**Likelihood Ratio**
LPS (pg/mL)					
128.5	63.0	44–78%	66.1	53–77%	1.86
132.5	63.0	44–78%	67.8	55–78%	1.96
142.5	63.0	44–78%	69.5	57–80%	2.06
149.5	63.0	44–78%	71.2	57–80%	2.19
151.0	59.3	41–75%	71.2	57–80%	2.06
155.0	55.6	37–72%	71.2	57–80%	1.93
169.0	48.2	31–66%	72.9	60–83%	1.78
BG (pg/mL)					
97.0	74.1	55–87%	81.7	70–89%	4.04
99.0	74.1	55–87%	83.3	72–91%	4.44
101.0	74.1	55–87%	85.0	74–92%	4.94
103.0	74.1	55–87%	90.0	80–95%	7.41
104.5	70.4	52–84%	90.0	80–95%	7.04
115.0	63.0	44–78%	90.0	80–95%	6.30
130.5	59.2	41–75%	91.7	82–96%	7.11

^1^ 95% confidence interval.

## Data Availability

The data are available from the corresponding author upon reasonable request.

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
