# Peer review of "Leaky Gut Syndrome Is Associated with Endotoxemia and Serum (1→3)-β-D-Glucan in Severe Dengue Infection"

_microorganisms, 2021, doi:10.3390/microorganisms9112390_

Round 1

Reviewer 1 Report

The manuscript “Leaky gut syndrome is associated with endotoxemia and serum (1→3)-β-D-glucan in severe dengue infection” by Chancharoenthana et. al. is a prospective observational study on dengue patients. A total of 48 and 38 patients were enrolled in febrile illness and critical phase, respectively, while 22 and 64 patients presented with or without the warning signs, respectively. Gut permeability was measured using lactulose-mannitol excretion ratio (LEMR). The association of serum lipopolysaccharide (LPS), LPS-binding protein (LBP), (1→3)-β-D-glucan 28 (BG), the indirect biomarkers for leaky gut, and inflammatory cytokines (IL-6, IL-8, IL-1β and TNF-α) with LEMR was analyzed. Positive LEMR test was found in 20 patients with warning signs in regardless phase of infection. In addition, seral levels of LPS, LBP and BG were increased in patients who developed severe dengue. Thus, the authors concluded that leaky gut syndrome might be associated with severity of dengue infection.

Vascular leakage is well known as one of the main characteristics of severe dengue. It is also known that increased serum LPS or endotoxin is correlated with dengue severity. Therefore, it is not surprised the authors found the association of endotoxemia in severe dengue infection. Thus, the significance and novelty of this manuscript are weak. Nevertheless, in this study, the authors found that, in addition to endotoxemia, BG (a major component of the fungal cell wall) was also increased, indicating gut barrier defects may contribute to dengue disease severity.

Author Response

We thank you the reviewer for the comments.

Reviewer 2 Report

The article by Chancharoenthana et al. suggests that leaky gut syndrome, which is assessed through lactulose-mannitol excretion ratio (LMER), may be indicative of or correlates with the severity of the disease in patients having dengue infection. Overall, the manuscript is well-written and delivers the idea clearly.

The following points should be addressed in the text:

Introduction: Regarding the statement about 40% of the world population being at risk:

  1. Dengue infection would be a more accurate work than dengue fever, as not all dengue patients may develop fever, as many infections are also asymptomatic.
  2. The authors may consider to update the values, as according to Messina et al. 2019, in 2080, 60% of the world population are estimated by projection studies to be at risk of dengue infection.

Line 144: days 1-3, the word days is missing.

Do the authors have possible explanation why BG does not present a high value as prediction marker, when both BG and LPS are biomarkers for gut barrier defects?

One additional limitation is that the study does not cover patients younger than 18 years old, which are an important age group prone to infection, as elderly patients. This should also be added.

Author Response

The article by Chancharoenthana et al. suggests that leaky gut syndrome, which is assessed through lactulose-mannitol excretion ratio (LMER), may be indicative of or correlates with the severity of the disease in patients having dengue infection. Overall, the manuscript is well-written and delivers the idea clearly.

The following points should be addressed in the text:

Introduction: Regarding the statement about 40% of the world population being at risk:

  1. Dengue infection would be a more accurate work than dengue fever, as not all dengue patients may develop fever, as many infections are also asymptomatic.
  2. The authors may consider to update the values, as according to Messina et al. 2019, in 2080, 60% of the world population are estimated by projection studies to be at risk of dengue infection.

ANS: We thank you the reviewer for suggestion. We have added in the revised manuscript.

Line 144: days 1-3, the word days is missing.

ANS: We have corrected accordingly.

Do the authors have possible explanation why BG does not present a high value as prediction marker, when both BG and LPS are biomarkers for gut barrier defects?

ANS: We thank you the reviewer for suggestion. We have added in the revised manuscript.

One additional limitation is that the study does not cover patients younger than 18 years old, which are an important age group prone to infection, as elderly patients. This should also be added.

ANS: We thank you the reviewer for suggestion. We have added the limitation in the revised manuscript.

This manuscript is a resubmission of an earlier submission. The following is a list of the peer review reports and author responses from that submission.